# ASPHALT II: Study Protocol for a Multi-Method Evaluation of a Comprehensive Peer-Led Youth Community Sport Programme Implemented in Low Resource Neighbourhoods

**DOI:** 10.3390/ijerph192215271

**Published:** 2022-11-18

**Authors:** Julie Hellesøe Christensen, Cecilie Karen Ljungmann, Charlotte Skau Pawlowski, Helene Rald Johnsen, Nikoline Olsen, Mathilde Hulgård, Adrian Bauman, Charlotte Demant Klinker

**Affiliations:** 1Health Promotion Research, Copenhagen University Hospital—Steno Diabetes Center Copenhagen, 2730 Herlev, Denmark; 2Research Unit for Active Living, Department of Sports Science and Clinical Biomechanics, University of Southern Denmark, 5230 Odense, Denmark; 3GAME, 2450 Copenhagen, Denmark; 4School of Public Health, Sydney University, Sydney, NSW 2006, Australia

**Keywords:** low resource neighborhoods, physical activity, prospective study, street sport, peer education

## Abstract

To reduce inequalities in children’s sport participation, studies are needed to explore ways in which children from low resource neighbourhoods can be engaged and retained in sport. GAME Community is a peer-led community sport programme which aims to promote physical activity through participation in inclusive street sports activities targeting 8–15-year-old children living in low resource neighbourhoods. The GAME Community intervention is implemented by the non-profit street sport organisation GAME. Five components support the implementation of GAME Community: (1) training of peer leaders; (2) a focus on inclusion of inactive girls; (3) parental involvement; (4) community engagement; (5) strengthened organisational support to peer leaders. In the ASPHALT II study, we aim to evaluate GAME Community and hence contribute to understanding how children growing up in low resource neighbourhoods can be engaged and retained in physical activity through participation in peer-led, community-based sport and to generate new understandings on the scale up of community interventions. The primary objective of the evaluation is to investigate the implementation of GAME Community and the programme’s reach, and to establish the functioning and mechanisms of the programme. The secondary objective is to establish the health enhancing potential of the programme. The evaluation of GAME Community involves three linked but independent studies that investigate (1) the functioning (i.e., mechanisms and processes), (2) reach, and (3) outcomes. The functioning of the five intervention components is investigated using qualitative methodologies. Programme reach will be investigated based on participant registrations. Systematic observations using a novel combination of validated tools will provide information on outcomes (physical activity level and social behaviour) during GAME Community activities. Investigating functioning, reach, and outcomes of the GAME Community intervention by using multiple methods is a strength, as different data complement and inform each other. This study will provide in-depth insights into if and how children living in low resource neighbourhoods can be engaged and retained in physical activity through participation in peer-led, community-based sport. Contributions of this evaluation include new understanding of the mechanisms and scalability of a community-based street sport intervention.

## 1. Introduction

Physical activity is important for children’s health in the short-term [1,2], as well as in the long-term, since physical activity behaviour tends to track from childhood and adolescence into adulthood [3]. Worldwide, only one out of five children adhere to international recommendations [4] and across high income countries there is inequality in physical activity with children from low socioeconomic status (SES) families being less physically active [5]. Increasing physical activity among children from low SES backgrounds is thus an important public health challenge [6], also contributing to reducing inequalities in health.

Different domains for children’s physical activity exist; recreation, transport, school, and household [7]. In the current study we will focus on recreational physical activity since participation in organised sport has proven to contribute to children’s physical activity [8] and further, sports club participation can be an important arena for children’s socialization experiences and social integration into the local community [9,10]. However, sports participation is influenced by the social environment in which children grow up, e.g., a recent systematic review and meta-analysis documented that the children living in low SES households participate less in sport [11]. Structural barriers to participation in organised sport for children from low SES families include: (i) fixed timetables for training, (ii) membership fees [9,12] and (iii) expectations of parent engagement [13]. Particularly girls from low SES families perceive barriers for participating in organised sport [12,14]. To redress inequality in sports participation among children, it is important to pay attention to alternatives to traditional sports club participation to attract and retain children from low SES families in being physically active [14,15].

Community-based health interventions are multicomponent interventions that combine individual and environmental change strategies across multiple settings aiming to promote well-being and health among population groups in a defined local community [16]. This intervention approach has the potential to effectively address community-level disparities [17]. Intervention strategies can include community-level influences on behaviour, such as local values and norms, and are more likely to generate local ownership and efficient use of resources [18]. Children spend considerable amounts of time in their community and community setting can include parents and other important influences to promote physical activity. These characteristics point towards the community setting as an important arena for promoting children’s physical activity [18].

The family also has an important influence on children’s sport participation [9,12], however, as children approach adolescence the influence of peers increases [19]. Studies have found that the social environment—being together with peers—significantly positively influences children’s physical activity behaviour [20,21]. Because of the strengthened peer influence in adolescence, peer education is a popular approach to health promotion with youth [22]. Further, peer-to-peer approaches may provide opportunities for reaching population groups who are usually considered harder to reach [23,24]. Peer-to-peer programmes build on the rationale that peer relations provide identification, credibility, trust, and role modelling influences [24]. Recent reviews have suggested that interventions based on a peer-to-peer approach may increase physical activity for children and adolescents, but more studies are needed to fully understand the potential of this approach in physical activity promotion [25,26].

To address inequality in sport participation among children in Denmark, the GAME Community programme has been developed through a partnership between the non-profit organisation GAME and research partners during 2018–2020 in the ASPHALT I study. GAME is a Danish-based street sport organisation with national as well as international activities in, Europe, Africa and the Middle East [27]. The overall aim of GAME is to offer accessible, and free-of-charge, peer-led street sport activities in low resource neighbourhoods targeting children living there. In ASPHALT I, the purpose was to improve and further develop GAME’s approach, mainly to improve stability of peer-led activities (e.g., that an activity continues over time and with few cancellations), increase retention of participants and peer-leaders (“Playmakers”), and enhance efforts to reach physically inactive groups, more specifically, physically inactive girls. A design process was conducted to better understand how to improve GAME’s existing programme. Researchers and staff from GAME participated in the process, which included fieldwork and the active involvement of girls, peer-leaders, parents, schoolteachers, and community stakeholders in the design process. The outcome of ASPHALT I was the GAME Community intervention, which is an expanded version of GAME’s previous programme. During the ASPHALT I study, it was established that it was not feasible to evaluate individual outcomes on participants in the context of GAME’s programme, as a pilot study did not generate sufficient information for this purpose. The pilot evaluation was abandoned as it was not possible to gain parental consent to evaluate individual outcomes on children as this contrasts with the GAME intervention in practice, which does not require parental consent.

The ASPHALT II study is a continued collaboration between research and practice, evaluating the functioning and outcomes of implementing the GAME Community intervention at scale in Denmark using multiple methods to overcome the challenges identified in ASPHALT I. Independent of research, GAME continues their engagement in neighbourhoods with existing GAME activities while gradually implementing additional GAME Community components. The scaling-up process consists of an increased number of components implemented in each setting rather than an increase in the number of settings targeted.

This paper presents the study protocol for the research conducted as part of the ASPHALT II study including a description and discussion of the intervention and the measurements to be used in the evaluation.

## 2. Materials and Methods

### 2.1. Study Aims

The overall aim of this study is to investigate how children living in low resource neighbourhoods can be engaged and retained in physical activity through participation in peer-led, community-based street sport. The primary objective is to investigate the implementation of GAME Community and the programme’s reach, and to establish the functioning and mechanisms of the programme. The secondary objective is to establish the health enhancing potential of the programme. The objectives are considered through three linked, but independent sub-studies described in detail further below.

### 2.2. Context

GAME is a non-profit organisation founded in 2002 with the objective of creating lasting social change through peer-led street sport and culture. One of GAME’s target areas is low resource neighbourhoods [28]. In Denmark, low resource neighbourhoods are defined based on employment rates, educational level, and criminal conviction rates among residents relative to the general population [29]. Street sport is characterised as sport activities, held in urban environments, of a non-commercial and non-professional nature (e.g., street basket, street soccer or street dance) [30]. The three main features of GAME’s programme are (1) a peer-to-peer approach in which local voluntary peer-leaders aged 16–25 years (Playmakers) are trained and supported in being role models and lead weekly diverse street sport activities for children aged 8–15 years (participants) living in low resource neighbourhoods. (2) Street sports activities are free of charge for children and do not require formal approvement by parents for children to participate. (3) The Playmakers are trained to conduct street sport activities that are inclusive and builds life skills and positive social relations among participants. In Denmark, GAME’s involvements range from neighbourhoods with one activity with few participating children to neighbourhoods with several weekly activities with large number of participants. A neighbourhood with GAME activities is defined as a GAME Zone, and thus a municipality with several low resource neighbourhoods may have several GAME Zones. Although some GAME Zones have been active for many years, GAME continuously seeks out possibilities for starting new GAME Zones. Further, some GAME Zones may be closed for a period or permanently, depending on a lack of recruitment of Playmakers, lack of facilities or lack of support from the neighbourhood. In May 2022, there were 105 educated and active peer leaders across 14 neighbourhoods (GAME Zones) in Denmark, responsible for a total of 28 weekly activities.

### 2.3. The GAME Community Intervention

GAME Community is a holistic, complex community intervention. The main component of the intervention is the weekly peer-led street sport activities, which take place at a set time and place in low resource neighbourhoods across Denmark. The implementation of these activities is supported by five synergistic intervention components: (1) training of Playmakers, (2) a focus on inclusion of inactive girls and specific activities for this purpose, (3) parental involvement, (4) community engagement, and (5) organisational support. The components, described in more detail below, will be integrated and implemented stepwise into all neighbourhoods with GAME activities during 2021–2023. The components will take different forms in different communities to match the local context. The implementation of GAME Community is dependent on partnership building in the local communities and attracting and retaining volunteer Playmakers. Therefore, local adaptation and a flexible approach is necessary in both the implementation and evaluation of GAME Community. The logic model underpinning GAME Community is shown in Figure 1. The logic model was developed during ASPHALT 1 in a collaboration between research and GAME.

#### 2.3.1. Training of Playmakers

The Playmaker training consists of an introductory course followed by four weekend camps during the following two years. Playmakers are enrolled continuously. Playmakers receive training to instruct and organise street sport activities for children within the neighbourhood. The training progress for each individual Playmaker who goes through the levels: basic, intermediate, advanced I, and advanced II, each level with corresponding learning objectives. Further, Playmakers receive tools they can use in the activities and in their everyday life with training in themes such as civic society engagement, gender equality, teamwork, life skills, conflict resolution and communication.

#### 2.3.2. Inclusion of Inactive Girls

The Playmaker training has a general focus on how to ensure an inclusive attitude and approach, particularly towards girls. Further, a group of female Playmakers are trained specifically to facilitate a new multisport activity called GAME Girl Zone. The GAME Girl Zone concept was co-developed during ASPHALT I to specifically target inactive girls and focusses less on competition and more on social relations and co-determination in the sport activities.

#### 2.3.3. Parental Involvement

To increase parent involvement and support, Playmakers and regional GAME officers participate in local social activities, e.g., BBQ-nights. Here, Playmakers and regional GAME officers introduce GAME street sport activities. Parents can meet Playmakers and discuss any potential concerns. Additionally, parents are invited to partake in national and regional events, such as GAME Finals. The programme will further collaborate with parents and parent groups who serve as informal authorities in their community to have them legitimise the Playmaker activities through their social networks, and to increase understanding in the community of the benefits of engaging in inclusive sport communities.

#### 2.3.4. Community Engagement

Before initiating activities in a neighbourhood, GAME establishes partnerships with local stakeholders, such as social housing development plans or local youth clubs. These partnerships are intended to increase involvement in the local community. The Playmakers are encouraged to participate as GAME representatives in community events or to facilitate events for the broader community with support from GAME. The purpose of such community events is to create awareness of local GAME activities to attract participants and ensure community awareness and support.

#### 2.3.5. Organisational Support

Playmakers, who often do not have any prior experience in volunteering work are supported locally by GAME zone coordinators to facilitate the overall functioning of the volunteer group and serve as role models to Playmakers. In addition, regional GAME officers function as contact people who support the Playmakers and follow up on training. These organisational structures are enforced to ensure that all Playmakers receive ongoing support and ensure that they feel confident in implementing weekly activities and avoid cancellations.

## 3. Evaluation Design and Methodology

The GAME Community programme is currently implemented at scale in Denmark by GAME but has not previously been subject to a research-based evaluation. Following Indig and colleagues’ [31] classification, the programme is thus ‘at scale dissemination’ without prior efficacy testing or real-world trial. With 20+ years of implementation, GAME’s activities have proven to be sustainable, but the health promotion outcomes are not known. With a primary focus on implementation outcomes and a secondary focus on intervention outcomes, the research design is a type 3 hybrid effectiveness-implementation design [32].

Embracing the structures and conditions of the GAME Community intervention, the evaluation design ensures that we will be able to describe the spirit of the intervention [33], determine both anticipated and unanticipated outcomes [34], and identify functioning of the programme (i.e., mechanisms and processes) [35,36,37]. As such, the overall methodology for our evaluation study is in line with recent suggestions calling for ‘holistic sense-making’ and prioritising generation of useful knowledge for practice in population health interventions [36,37]. Further, the evaluation is situated within the theory-based research paradigm for evaluating complex interventions as this paradigm is concerned with evaluating how change is brought about, embracing the interplay between mechanisms and context, which is in line with the aim of our study. We apply quantitative, qualitative, observational, and ethnographic methods to determine ’What happened’ [36] in the implementation of GAME Community. A summary of the proposed interlinked, yet independent, sub-studies that guide the evaluation is provided in Table 1.

### 3.1. Sub-Study 1: Functioning of the GAME Community Programme

We assess the level of implementation of the five GAME Community components in all GAME Zones (see Section 3.2) and their function during implementation in six neighbourhoods with high/low implementation. By comparing implementation data related to perceptions of intervention delivery in context with data on reach and retention, we investigate relationships between implementation determinants and implementation outcomes [38] to understand the functioning of the intervention.

Based on an implementation scoring of all GAME zones conducted in Fall 2022 (see Section 3.2), three GAME Zones with lowest score and three GAME Zones with highest score will be selected for a qualitative study of the functioning of the intervention. We expect this maximum variation approach to provide an information-rich comparison [39] for exploring implementation of the components and their perceived impact. We will use qualitative methods to gain insight into ways in which the five components are implemented, adapted locally, interact, and how the programme and the context into which it is implemented are mutually influenced in the process. One-on-one interviews will be conducted in 2023 with local stakeholders who are familiar with GAME Community. Moreover, focus group interviews will be conducted with Playmakers and networks representing local parents. The interviews will explore how the GAME Community elements have been implemented locally and what are the perceived outcomes of the implementation. An experienced researcher will lead the interviews following a semi-structured interview guide developed to standardize interviews between participants. To enhance trustworthiness, two researchers will code the interview data independently and reach consensus through discussion of discrepancies.

### 3.2. Sub-Study 2: Reach and Retention of Participants

For all GAME Zones, the degree of implementation of the GAME Community components will be assessed three times (in Fall 2021, 2022, and 2023), as the level of implementation at each GAME Zone is expected to either deteriorate or improve over time. Both scenarios are possible, although the intention is to achieve an increased implementation over time in all GAME Zones. Implementation degree is assessed by interviewing the regional GAME officers who oversee and support the implementation of GAME’s activities across all zones in Denmark. In 2021, comprehensive descriptions of the five GAME Community components were developed in collaboration between GAME and researchers. Based on these descriptions, the research team developed a structured interview guide to assess the implementation of the five GAME community components. For each GAME Zone, each component will be assigned a score from 1 (not implemented) to 5 (fully implemented). All GAME Zones will thus achieve a combined score between 5 and 25. By comparing GAME Zones with high and low implementation, we will explore the relationship between implementation of various programme components and programme reach in different contexts.

Simple registrations of all participants attending GAME activities across all GAME Zones will be collected from April 2022 using an app developed, managed, and owned by GAME. When a child attends an activity for the first time, the Playmaker will register the child’s attendance (activity and zone ID and date) in the app together with background information such as name, surname initials, gender, age, and current participation in organised leisure activities (assessed as current participation in sport, participation in other activities, or no current participation in organised activities). At all forthcoming GAME activities, recurring and new participant attendance will be registered. The app allows data to be linked if the same child attends different activities. Data from the app will handed over to research as an anonymised dataset. The participant registration data will allow for exploration of programme reach and retention of participants, including stratified analyses by, for example, current participation in organised leisure activities at first attendance. Moreover, registrations will provide insight into the number of activities and stability of activities over time.

To understand relations between implementation of the GAME Community components and programme reach, retention and number and stability of activities, we will investigate the functioning of the intervention by considering the role of each component in the change process [35,40]. The participant registrations related to programme reach will further be combined with qualitative data of the perceived quality of the implementation and the functioning of the programme elements. These data will be collected as an integral part of the interviews conducted during Sub-study 1 (See Section 3.1). This will provide insight into the ways in which the GAME Community elements and contextual factors attract and retain participants, including girls who did not participate in sport prior to GAME participation.

The programme’s ability to attract and retain physically inactive girls will be explored in a qualitative study utilising ethnographic approaches. Girls participating in GAME activities will be followed by participant observation and interviewed during 2023. Girls will be recruited from the three GAME Zones with the highest participation of girls to provide insights on girls’ experiences of participating in GAME Zone activities, social interactions, and interaction with Playmakers.

### 3.3. Sub-Study 3: The Health Enhancing Potential of GAME Community

To have a health promoting effect, participation by children and youth should be continuous, and activities should be characterised by a moderate to vigorous activity level and/or pro-social behaviours. To investigate whether GAME Community activities have these characteristics, an observational study will be conducted. Data will be collected using structured observations across all activities that comply with being in a public open space and with a mean of at least five participants to ensure anonymity. Data on each of GAME’s activities that adhere to this standard will be collected at four time points: twice during spring 2022 and twice during fall 2022. We will use a combination of the validated systematic observation procedures SOFIT (System for Observing Fitness Instruction Time) [41] and SOCARP (System for Observing Children’s Activity and Relationships during Play) [42], both of which use momentary time sampling as an observation approach. SOFIT was developed for a physical education context while SOCARP was developed for observing playground settings. The two procedures use a similar observation technique and have several similar observation elements (e.g., physical activity level and activity content) and can therefore be used in combination. To maintain a manageable instrument, we will select the observation elements from each protocol which are most pertinent to our aim of assessing physical activity level and social interactions at the activity level. Using SOFIT and SOCARP, we will assess (a) physical activity level among participating children, (b) participants’ social behaviour and interactions, (c) GAME activity context and (d) features of the physical context, e.g., accessibility and amenities. An estimated total of 15–20 weekly GAME activities will be observed four times. Observers will be trained in accordance with SOFIT and SOCARP protocols [43,44]. Findings from these systematic observations will be analysed to characterise GAME activities in terms of their physical activity content and the children’s physical activity level and the degree to which they are characterised by pro-social behaviour. Observation data will be compared with participant registrations (see Section 3.2) from the included GAME Zones to explore patterns in activity characteristics and to explore characteristics of participants who are reached and retained.

## 4. Discussion

This study protocol presents how we will evaluate the functioning, reach, and outcomes of the GAME Community intervention using a multi-method design including three linked but independent studies. The results will provide unique insights into how and if children living in low resource neighbourhoods can be engaged and retained in physical activity through participation in peer-led, community-based sport.

### 4.1. Importance of This Study

It is important to evaluate the health enhancing potentials of practice-based interventions to determine if scarce societal resources are to be prioritised towards an intervention. In the case of GAME Community this is challenging, as individual level health outcomes cannot easily be assessed. Instead, we propose to evaluate intermediate health outcomes [45] of GAME Community (in terms of physical activity level and social behaviour and interactions) at an aggregated activity level using systematic observations. Further, we will synthesise data to evaluate implementation outcomes [38] in terms of reach (number of previously inactive children attracted, total number of children that participate and their retention) at an individual level using anonymised participant registration. Using this novel combination, the results will provide an indication of the health enhancing potential of the programme by establishing whether the participating children are attracted and retained in activities characterised by physical activity and positive social relations.

GAME Community is a complex health promotion intervention characterised by several interacting components that are delivered on several levels [37,46]. Further, for Playmakers and trainings to be successful, GAME Community needs to be implemented in a supportive environment. As such, context is part of the essence of GAME Community, with the programme both *relying on* [… a positive and supportive…] *context* to be a success but at the same time also *altering the context* by forming new partnerships between stakeholders in the neighbourhood and forming relationships among parents [36]. In sum, it is neither possible nor relevant to deconstruct a health promotion intervention such as GAME Community into its components and establish causal relationships between each component and behavioural or health outcomes, and an evaluation and research design that focuses only on internal validity, such as randomised trials, is ill-suited as a research framework in this community-based study. With an increased awareness of the pitfalls of applying strict controlled scientific study designs in evaluating complex public health interventions, recognition is growing that studies that engage with questions beyond the efficacy and effectiveness of complex interventions can provide important insights for research and practice [32,37]. With this in mind, this study protocol presents research initiated to evaluate the implementation, outcomes, reach, and functioning of a peer-led, community-based street sport intervention implemented at scale in low resource neighbourhoods, with a particular lens looking at girls.

### 4.2. Strengths and Limitations

The study was designed in response to the real-world conditions in which the programme is implemented. The programme is an equity focused intervention, designed for a low physically active, socially disadvantaged group who may not have access to organised sport, active transport, and other modes of physical activity. It is only one component of total physical activity [7], but in this population, it may be the only physical activity they participate in.

Throughout the planning of the research design, practical issues, important knowledge gaps, and relevant outcome measures have been discussed with GAME. GAME will assist in recruiting participants for the study and the study design will continuously be discussed with GAME to ensure that the design and content is practical possible and will generate knowledge of relevance to both research and practice. The close collaboration with professionals from GAME in the description and planning of the study is evident in the co-development of this protocol. Other studies have shown that when design and planning experiences are shared between professions engaged in the study, it is more likely that the partners will learn and generate valuable ideas, increase quality and flexibility, improve efficiency, and simulate appropriate use of resources [47]. A particular strength of the study is that it was planned and developed as an equal partnership between practitioners and researchers, each using their core competences to share ideas and jointly complete all tasks in the study. The close collaboration between research and practice may also increase the likelihood of sustained implementation [48].

Investigating functioning, reach, and outcomes of the GAME Community intervention by using a combination of quantitative and qualitative methods is a strength, as the different types of data complement each other and results of one part of the study will inform the subsequent parts [49]. Thus, the study will provide in-depth insights in how children growing up in low resource neighbourhoods can be engaged and retained in physical activity through participation in peer-led, community-based sport. A novel aspect of our study is the combination of using SOCARP and SOFIT to determine the participating children’s PA level, social behaviour, interactions, activity content and context. Previously, several studies have used systematic observations to evaluate physical activity intervention studies [50]. To our knowledge, using systematic SOCARP and SOFIT observations in combination has not been used before in longitudinal studies evaluating physical activity interventions in neighbourhoods. Using systematic observation in our study is limited to activities in public open spaces while we cannot obtain informed consent from parents, as children show up on the day and are not registered. Therefore, the study design for the SOFIT/SOCARP observations has been adapted to only include activities that take place in public open spaces and thereby excludes all indoor activities, for example street dance activities. Further, since the activities rely on Playmakers’ and participants’ engagement, a concern is if activities will be cancelled and thus, if we can reach an appropriate number of observations for statistical sub analysis. Another limitation is the assessment of implementation degree based on GAME’s own subjective appraisal of how the GAME Community components have been implemented in each GAME Zone. To minimize this bias two people instead of one are continuously interviewed as they can discus and verify each other’s assessments during interviews. Additionally, the two people recruited for interview are regional GAME officers probably being more objective in their assessment than local stakeholders.

## 5. Conclusions

The ASPHALT II study is an evaluation of the ongoing GAME Community intervention targeting children living in low resource neighborhoods with peer-led street sport activities. This paper presents the intervention and how we will assess the program’s reach, functioning, mechanisms, and health enhancing potential.

The ASPHALT II study provides a creative and novel approach in evaluating an intervention that is already being implemented at scale. The use of multiple methods, quantitative longitudinal participant registrations, direct observations, ethnographic field study, focus groups and interviews will provide a comprehensive understanding of the outcomes of the GAME Community intervention across multiple perspectives. The results from our study can inform the potential scale up of other community-based interventions in low resource neighbourhoods in the future.

## Figures and Tables

**Figure 1 ijerph-19-15271-f001:**
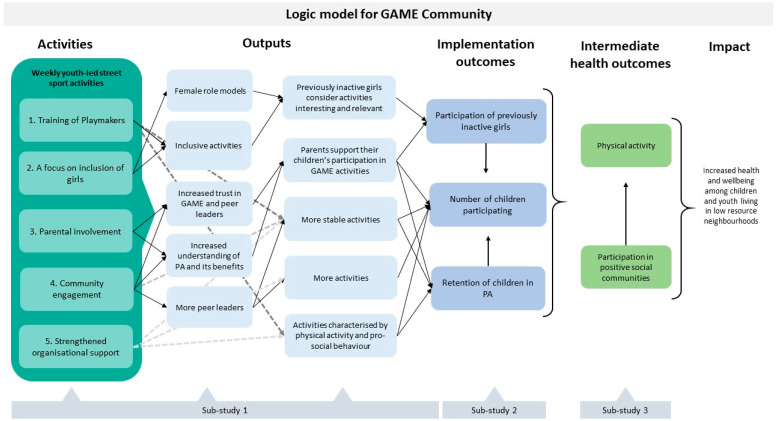
Logic model for the GAME Community Intervention.

**Table 1 ijerph-19-15271-t001:** Sub-studies in the ASPHALT II evaluation.

Sub-Study	Research Question	Design and Methods	Population	Data	Time
**1. Functioning of the intervention *(activities and outputs)***	What is the functioning of the GAME Community programme?	Qualitative interviews and focus groups	Playmakers, parents, community stakeholders from 6 GAME Zones with varying implementation scores	Participants’ perceptions of the implementation and impact of the five GAME community components	2023
**2. Reach and retention of participants *(implementation outcomes)***	What is the relationship between degree of implementation and implementation outcomes?	Structured interviews with regional GAME officers and longitudinal participant registration	All GAME Zones	Each of the five GAME community components will be assigned a score from 1 (not implemented) to 5 (fully implemented) and combined with app data	Time 1: 2021, Time 2: 2022, Time 3: 2023
	What are the relations between implementation of the GAME Community components and programme reach and retention?	Qualitative interviews, focus groups and longitudinal participant registration	Playmakers, parents, community stakeholders from 6 GAME Zones with varying implementation scores	Participants’ perceptions of the implementation combined with participant registrations (app) on: gender, age, participation in other leisure activities, number of GAME activities attended	2022–2023
	Which components of the programme contribute to engage and retain girls in GAME activities?	Ethnographic field study	Girls who are active in GAME Zones with a high degree of girl participation	Participating girls’ experiences of participating in GAME activities, social interactions and interaction with peer leaders	2023
**3. The health enhancing potential of GAME Community *(intermediate health outcomes)***	What is the health enhancing potential of participation in GAME Community? E.g., retention in activities characterised by physical activity and pro-social behaviour?	Systematic observations of each activity at four time points using a combination of SOFIT and SOCARP. Longitudinal participant registration	All children attending GAME activities that are conducted in public open space and with a mean attendance of >5 children	(a) PA level among participating children, (b) participants’ social behaviour and interactions, (c) activity content and (d) features of the physical context will be assessed and combined with app data.	2022–2023

## Data Availability

Not applicable.

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
