# Peer review of "ASPHALT II: Study Protocol for a Multi-Method Evaluation of a Comprehensive Peer-Led Youth Community Sport Programme Implemented in Low Resource Neighbourhoods"

_ijerph, 2022, doi:10.3390/ijerph192215271_

Round 1

Reviewer 1 Report

Extensive Engli is required

Please elaborate more what is meant by "Community interventions  " and make references to to the relevant literature. Make more specific references also to the role of the social environment on youngsters' behaviors.

Much more than that, the paper does not present any evidence; it is just a presentation of what is going to be done in order to assess/evaluate the results of a specific project.

In addition, the paper does not refer to any relevant theory or methodology

I think it would be better to present the findings of the final evaluation. At the moment, the paper has nothing to offer

Author Response

Thank you for taking your time to review our paper. Please find below a point-by-point response to your comments. Your comments are included as well as how we have dealt with your comment.

Extensive Engli is required

  • The native English speaker and co-author, Professor Adrian Bauman, has edited the English language and style in the paper and we hope it meets the standards

Please elaborate more what is meant by "Community interventions " and make references to the relevant literature. Make more specific references also to the role of the social environment on youngsters' behaviors.

  • Community interventions as well as community-based interventions are well-used terms. We have now included a definition and references on that in the introduction section lines 65-68. In the introduction we have further added the following sentence and references in lines 76-78: Studies have found that the social environment – being together with peers positively influences children’s physical activity behaviour.

Much more than that, the paper does not present any evidence; it is just a presentation of what is going to be done in order to assess/evaluate the results of a specific project.

  • . As this is a study protocol paper, the aim is not to present results of the project but to present the evaluation design, content, and methodologies that we are going to use in the evaluation of the project. Later papers will describe the results and evidence, using the framework provided here.

In addition, the paper does not refer to any relevant theory or methodology

  • In section 3. ‘Evaluation framework’ we present the methods we will apply to collect the proper evidence to evaluate the program. Table 1 presents a summary of the proposed interlinked, yet independent, sub-studies that guide the evaluation with the column headed Design describing the methodology applied in each of the sub-studies. This is further expanded in the following sub-sections where the methodology is described in more detail. We agree that the heading ‘Design’ does not fully describe the content and we have now changed it to ‘Design and methods’ to accommodate your comment.
  • The theory used is the 'theory of complex programs', so no single theory is possible; It uses a social determinants framework and a systems approach, but they are not specific theories. To clarify this, we have now added in lines 238-241: The evaluation is situated within the Theory-based research paradigm for evaluating complex interventions as this paradigm is concerned with evaluating how change is brought about, embracing the interplay between mechanisms and context, which is in line with the aim of our study.

I think it would be better to present the findings of the final evaluation. At the moment, the paper has nothing to offer

  • Per definition a study protocol is an outline of the steps research will undertake to evaluate a program or project, including objective, purpose, procedures, and methods, as is also the content of our paper. The findings of the different sub-studies will be published elsewhere in the peer reviewed literature.

Reviewer 2 Report

The authors should be commended for their effort of investigating how to reduce inequalities in children’s sport participation. Overall this article was well written and organized in my opinion. I do not have much feedback to offer except for the following:

Authors should consider adding a literature review section on previous research that examine inequality in sport participation.

Could you provide more background on ASPHALT study?

Could you provide more details of your methods? For instance, regarding the qualitative method, authors should provide further detail on interview questions, structure of the interview, and information of interviewees. How many researchers are involved in the coding process? What strategies have been taken to establish validity and reliability in qualitative analysis?

Finally a statement about limitations of the study needs to be made.

Author Response

Thank you for taking your time to review our paper. Please find below a point-by-point response to your comments. Your comments are included as well as how we have dealt with your comment.

The authors should be commended for their effort of investigating how to reduce inequalities in children’s sport participation. Overall this article was well written and organized in my opinion. I do not have much feedback to offer except for the following:

  • Thank you for your overall positive assessment of our study protocol. We are glad that you appreciate our efforts to produce new, real-world evidence on ways to reduce inequalities in children’s sport participation

Authors should consider adding a literature review section on previous research that examine inequality in sport participation.

  • Thank you for this comment. We have now in the introduction clarified that our claim that inequality in sports participation exist builds on a systematic review and meta-analyses on this matter; line 56-60 now reads: However, sports participation is influenced by the social environment in which children grow up, e.g., a recent systematic review and meta-analysis documented that the children living in low SES households participate less in sport (ref to Fair play? Participation equity in organised sport and physical activity among children and adolescents in high income countries: a systematic review and meta-analysis)

Could you provide more background on ASPHALT study?

  • We believe we have given quite rich information to understand the aim, scope and outcomes of the ASPHALT 1 study, which we think the reviewer is referring to, please see below, or perhaps specify what more information might be needed: In ASPHALT I, the purpose was to improve and further develop GAME’s approach, mainly to improve stability of peer-led activities (e.g., that an activity continues over time and with few cancellations), increase retention of participants and peer-leaders (“Play-makers”), and enhance efforts to reach physically inactive groups, more specifically, physically inactive girls. A design process was conducted to better understand how to improve GAME’s existing programme. Researchers and staff from GAME participated in the process, which included fieldwork and the active involvement of girls, peer-leaders, parents, schoolteachers, and community stakeholders in the design process. The outcome of ASPHALT I was the GAME Community intervention, which is an expanded version of GAME’s previous programme. During the ASPHALT I study, it was established that it was not feasible to evaluate individual outcomes on participants in the context of GAME’s programme, as a pilot study did not generate sufficient information for this purpose. The pilot evaluation was abandoned as it was not possible to gain parental consent to evaluate individual outcomes on children as this contrasts with the GAME intervention in practice, which does not require parental consent (lines 94-109).

Could you provide more details of your methods? For instance, regarding the qualitative method, authors should provide further detail on interview questions, structure of the interview, and information of interviewees. How many researchers are involved in the coding process? What strategies have been taken to establish validity and reliability in qualitative analysis?

  • Thank you for this comment. In a study protocol it is always a struggle to provide enough information to illustrate the content and scope of the suggested research keeping in mind the length of the paper and the flexibility in letting the context and data talk. However, to accommodates your comment, we have now added the following supplement information in lines 268-272: An experienced researcher will lead the interviews following a semi-structured interview guide developed to standardize interviews between participants. To enhance trustworthiness, two researchers will code the interview data independently and reach consensus through discussion of discrepancies.

Finally, a statement about limitations of the study needs to be made.

  • In the strengths and limitations section we have added more limitations to the already described limitations. We have added following sentence lines 390-395: This programme is an equity focused intervention, designed for a low physically active, socially disadvantaged group who may not have access to organised sport, active transport, and other modes of physical activity. It is only one component of total physical activity, but in this population, it may be the only physical activity they participate in. Further, in relation to the limitation on observations we have added following sentence lines 425-427: Further, since the activities relay on Playmakers’ and participants’ engagement, a concern is, if activities will be cancelled and thus, if we can reach an appropriate number of observations for statistical sub analysis.

Reviewer 3 Report

Abstract- spell out acronyms (e.g., what does GAME or SOFIT stand for?)

Introduction- what about the role of schools? Children spend a lot of time in school, how does this relate to their physical activity behaviors or potentially support community-based measures? This is an important domain of physical activity and should be mentioned as a limitation.

The introduction should include some discussion of the components of physical activity for children, such as active transportation, leisure time, physical education class, etc.

Is there an assessment of the built environment of where the program is implemented in different neighborhoods? Such as, accessibility of play spaces, size or amenities, or safety? This may explain some of the variation in participation in the program between neighborhoods. If not, this may be a limitation of this study.

Also, I wonder if there is a component to ask for the peer leaders about how to improve attendance / increase physical activity in their neighborhoods?

Another discussion point is to consider this program as one component in the overall levels of physical activity.

Author Response

Abstract- spell out acronyms (e.g., what does GAME or SOFIT stand for?)

  • We have adjusted the abstract to accommodate this comment. In the abstract we now clarify that GAME is organization to help the reader understand that it is not an acronym and we have rephrased the sentence on our observation tools to avoid the use of SOFIT and SOCARP as we believe this would lengthen the abstract unnecessarily . I now reads: The GAME Community intervention is implemented by the non-profit street sport organisation GAME (line 18-19). AND: Systematic observations using a novel combination of validated tools will provide information on outcomes (physical activity level and social behaviour) during GAME Community activities (31-34)

Introduction- what about the role of schools? Children spend a lot of time in school, how does this relate to their physical activity behaviors or potentially support community-based measures? This is an important domain of physical activity and should be mentioned as a limitation. The introduction should include some discussion of the components of physical activity for children, such as active transportation, leisure time, physical education class, etc.

  • We agree that the school is one important domain for physical activity as well as the other domains mentioned. The physical activity investigated in this study is not PE classes, not active transportation, and not leisure time usual PA. The innovation here is addressing adolescent culture and context with a new way of delivering and understanding physical activity programs for this group. However, we have introduced to reader into the different PA domains and then zoomed into the recreational domain we have focused on and describe why we believe that a community-based intervention is an important approach among low SES populations. In the discussion we have also followed up on this (see our answer to your comment below). The introduction now reads: Different domains for children’s physical activity exist; recreation, transport, school, and household (ref.). In the current study we will focus on recreational physical activity since participation in organised sport has proven to contribute to children’s physical activity [7] and further, sports club participation can be an important arena for children’s socialization experiences and social integration into the local community [8,9]. (line 52-57)

Is there an assessment of the built environment of where the program is implemented in different neighborhoods? Such as, accessibility of play spaces, size or amenities, or safety? This may explain some of the variation in participation in the program between neighborhoods. If not, this may be a limitation of this study.

  • Thank you for this comment – and yes, this is assessed as component “d” of the systematic observation tools, however, we can now see that this component is only mentioned in table 1, sub-studie 3, the Data collumn and we have now also included a short description on the actual content in section 3.3., lines (340-342): Using SOFIT and SOCARP, we will assess a) physical activity level among participating children, b) participants’ social behaviour and interactions, and c) GAME activity context and d) features of the physical context, e.g., accessibility and amenities.
  • Also, features of the built environment will be explored during the qualitative components as these will also explore the context as described in sub-section 3.1. line 262-265: We will use qualitative methods to gain insight into ways in which the five components are implemented, adapted locally, interact, and how the programme and the context into which it is implemented are mutually influenced in the process

Also, I wonder if there is a component to ask for the peer leaders about how to improve attendance / increase physical activity in their neighborhoods?

Focus group interviews will be conducted with Playmakers and the interviews will explore how the GAME Community elements have been implemented locally and what the perceived outcomes of the implementation is (line 267-270). As improved attendance and increase in physical activity are intended outcomes of the intervention (see for instance Figure 1. Logic model for the GAME Community Intervention) these components are integrated into the interviews with the peer leaders and local parents. Another discussion point is to consider this program as one component in the overall levels of physical activity.

  • In the discussion we have added the following lines 412-416: This programme is an equity focused intervention, designed for a low physically active, socially disadvantaged group who may not have access to organised sport, active transport, and other modes of physical activity. It is only one component of total physical activity, but in this population, it may be the only physical activity they participate in.

Round 2

Reviewer 1 Report

The paper is a quite early stage; it does not present any data or evidences. It is a presentation of how a project is going to be assessed. At the moment there are no relevant  information or data to be discussed and disseminated. Authors should writer a paper after the collection and analyses of the data